# Sensor Transformation Attention Networks

## Abstract

Recent work on encoder-decoder models for sequence-to-sequence mapping has shown that integrating both temporal and spatial attentional mechanisms into neural networks increases the performance of the system substantially. We report on a new modular network architecture that applies an attentional mechanism not on temporal and spatial regions of the input, but on sensor selection for multi-sensor setups. This network called the sensor transformation attention network (STAN) is evaluated in scenarios which include the presence of natural noise or synthetic dynamic noise. We demonstrate how the attentional signal responds dynamically to changing noise levels and sensor-specific noise, leading to reduced word error rates (WERs) on both audio and visual tasks using TIDIGITS and GRID; and also on CHiME-3, a multi-microphone real-world noisy dataset. The improvement grows as more channels are corrupted as demonstrated on the CHiME-3 dataset. Moreover, the proposed STAN architecture naturally introduces a number of advantages including ease of removing sensors from existing architectures, attentional interpretability, and increased robustness to a variety of noise environments.

## 1 Introduction and Motivation

Attentional mechanisms have shown improved performance as part of the encoder-decoder based sequence-to-sequence framework for applications such as image captioning (Xu et al., 2015), speech recognition (Bahdanau et al., 2016; Chan et al., 2016), and machine translation (Bahdanau et al., 2014; Wu et al., 2016). Dynamic and shifting attention, for example, on salient attributes within an image helps in image captioning as demonstrated by the state-of-art results on multiple benchmark datasets (Xu et al., 2015). Similarly, an attention-based recurrent sequence generator network can replace the Hidden Markov Model (HMM) typically used in a large vocabulary continuous speech recognition system, allowing an HMM-free RNN-based network to be trained for end-to-end speech recognition (Bahdanau et al., 2016).

While attentional mechanisms have mostly been applied to both spatial and temporal features, this work focuses on attention used in *sensor selection*. We introduce the STAN architecture that embeds an attentional mechanism for sensor selection and supports multi-sensor as well as multi-modal inputs. This attentional mechanism allows STANs to dynamically focus on sensors with higher signal-to-noise ratio (SNR) and its output is highly interpretable. Because of their inherently modular architecture, STANs remain operational even when sensors are removed after training. The same modularity makes STANs attractive for tasks that make use of multi-sensor integration. The STAN architecture can be seen as a generalization of existing less-modular network types that include attention in multi-sensor setups (Kim & Lane, 2016; Hori et al., 2017).

This work consists of three main sections. First, we formally introduce the STAN architecture in section 2. In the first evaluation phase in section 3, we demonstrate the proper function of the attentional mechanism in synthetic noise environments with multiple audio sensors (TIDIGITS dataset) and multiple video sensors (GRID). The second evaluation phase in section 4 covers the multi-microphone real world dataset CHiME-3, where we show the proper functioning of the STAN attentional mechanism on natural noise and the robustness of STANs with respect to altered sensor configurations.

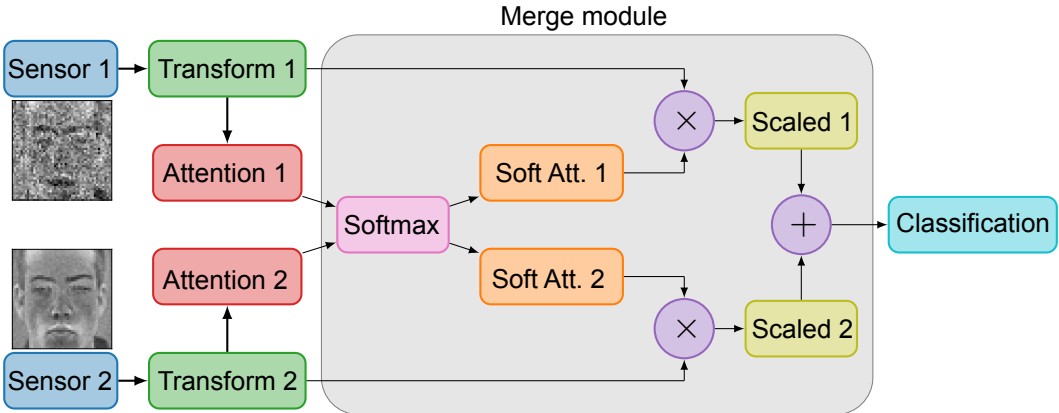

Figure 1: STAN model architecture for a setup with two video sensors.

## 2 SENSOR TRANSFORMATION ATTENTION NETWORK

We introduce the STANs in Figure 1 as a general network architecture that can be described with five elementary building blocks: (1) input sensors, (2) transformation modules, (3) attention modules, (4) a sensor merge module and (5) a classification module.

Formally, we introduce a pool of $N$ sensors $s^i$ with $i = 1, ..., N$. We assume working with time series that are binned into $k = 1, ..., K$ frames, and that each sensor $s^i$ provides a feature vector $f_k^i$ for frame $k$. The transformation modules transform the feature vectors $f_k^i$ to the transformed feature vectors $t_k^i$. If no transformation is desired, we maintain the identity $f_k^i = t_k^i$. The attention modules compute the scalar attention score $z_k^i$ for their corresponding input $t_k^i$. The sensor merge module computes the attention weights $a_k^i$ by performing a softmax operation over all attention scores $z_k^i \in \{z_k^1, ..., z_k^N\}$ at a frame $k$ (Equation 1). Each transformed feature vector $t_k^i$ is then scaled by the corresponding attention weight $a_k^i$ and merged by an adding operation (Equation 2). The resulting - transformed, scaled and merged - feature vectors $t_k^{\text{merged}}$ are then presented to the classification module for classification.

$$a_k^i(z) = \frac{\exp(z_k^i)}{\sum\limits_{j=1}^{N} \exp(z_k^j)} \qquad (1) \qquad\qquad t_k^{\text{merged}} = \sum_{i=1}^{N} a_k^i \cdot t_k^i \qquad (2)$$

The focus of this work is sequence-to-sequence mapping on time-series in which the attention values are computed on a per-frame basis. This allows the STAN architecture to dynamically adjust and compensate for changes in signal quality due to noise, sensor failure, or informativeness. As the attention modules are required to study the dynamics of the input stream to determine signal quality, recurrent neural networks (RNNs) such as gated recurrent units (GRUs) (Cho et al., 2014) or long short-term memory (LSTM) cells (Hochreiter & Schmidhuber, 1997) seem a natural choice. The transformation layers heavily depend on the input modality, with densely connected units or RNN variants being a good choice for audio features and convolutional neural networks (CNNs) (LeCun et al., 1998; Krizhevsky et al., 2012) well adapted for images/videos.

## 3 EXPERIMENTS WITH SYNTHETIC NOISE

This section presents experiments that show the performance of STANs in environments with dynamically changing noise levels of a synthetic noise source. The experiments are carried out for multiple audio sensors (TIDIGITS dataset) and multiple video sensors (GRID dataset). Both of these datasets do not contain multiple sensors of the same modality, so the available sensor was cloned to establish a multi-sensor scenario. With such a pool of identical sensors, further incentive is necessary in order to learn the attentional mechanism. In the following synthetic noise experiments, every sensor is corrupted with a different Gaussian noise source whose noise levels change over time. Consequently,

Table 1: Models for the TIDIGITS evaluation, with 'G'=GRU and 'D'=dense unit

| Name | Architecture | Audio sensors | Transformation Module | Attention Module | Classification Module | Parameters |
|------|-------------|---------------|----------------------|------------------|----------------------|-----------|
| Single Audio | Baseline | 1 | Identity | None | (150,100,12)-G,G,D | 162K |
| Double Audio STAN | STAN | 2 | Identity | (20,1)-G,D | (150,100,12)-G,G,D | 170K |
| Triple Audio STAN | STAN | 3 | Identity | (20,1)-G,D | (150,100,12)-G,G,D | 173K |
| Double Audio Concat | Concatenation | 2 | Identity | None | (150,100,12)-G,G,D | 180K |
| Triple Audio Concat | Concatenation | 3 | Identity | None | (150,100,12)-G,G,D | 197K |

the output of a particular sensor might temporarily be too noisy for a network to solve a given task well, therefore, attending to less noisy sensors is beneficial. We refer to this noise type as random walk noise and elaborate our implementation in Appendix A. The actual noise generation process is rather arbitrary, but sufficient if the noise level varies enough to learn the attentional mechanism during training. In all synthetic noise experiments, a Gaussian noise source with $\mu = 0$ and $\sigma$ between $[0, 3]$ was added to zero-mean and unit-variance normalized samples during training. We further refer to $\sigma$ as the noise level.

### 3.1 SYNTHETIC NOISE ON AUDIO

**Dataset** The TIDIGITS dataset (Leonard & Doddington, 1993) was used as an initial evaluation task to demonstrate the response of the attentional signal to different levels of noise in multiple sensors. The dataset consists of 11 spoken digits ('oh', 'zero' and '1' to '9') in sequences of 1 to 7 digits in length, e.g '1-3-7' or '5-4-9-9-8'. The raw audio data was converted to 39-dimensional Mel-frequency cepstral coefficients (MFCCs) (Davis & Mermelstein, 1980) using a frame size of 25ms and a frame shift of 10ms. The features are zero-mean and unit-variance normalized on the whole dataset. The WER is used as a performance metric, with single digits considered as words.

**Models** A total of five models were evaluated, with a summary given in Table 1. The STAN models use separate parameters for each sensor's attention module. A leaky ReLU non-linearity (Maas et al., 2013) is applied on the dense unit of the attention modules. In order to evaluate the potential benefit of STAN architectures, they are compared against two simpler sensor concatenation models, which concatenate all sensors into a single representation that is presented to the classification module.

**Training** In order to automatically learn the alignments between speech frames and label sequences, the Connectionist Temporal Classification (CTC) (Graves et al., 2006) objective was adopted. All models were trained with the ADAM optimizer (Kingma & Ba, 2014) for a maximum of 100 epochs, with early stopping preventing overfitting.

**Results** Two key results emerge from this initial experiment: firstly, the attentional mechanism generalizes across a variety of noise types; and secondly, STAN models outperform, in error rate, merely concatenating input features together. To demonstrate, Figure 2 (top left) shows the attention response of a Double Audio STAN model with two audio sensors in response to random walk noise. A sample from the test set was corrupted by the default random walk noise. The model shows the desired negative correlation between noise level and attention: when the noise level for a sensor goes up, the attention given to the same sensor goes down. Furthermore, without additional training, the same model is evaluated against novel noise types in Figure 2 (left). The attention modules successfully focus their attention on the sensor with the lowest noise level under a variety of noise conditions. In situations where the noise level of both sensors is low, such as seen in the burst noise case, the attention settles in an equilibrium between both sensors.

To determine whether the attention across sensors actually improves performance, the STAN models are evaluated against a baseline single sensor model and concatenation models under both clean and noisy conditions. With the clean test set, all available sensors are presented the same clean signal. With the noisy test set, each sensor's data is corrupted by unique random walk noise with a noise level between $[0, 3\sigma]$. The results are reported in Figure 3(a). All models achieve comparably low WER around 1% on the clean test set, despite training on noisy conditions, implying the STAN architecture does not have negative implications for clean signals. On the noisy test set, the two- and three-sensor STAN models perform better, with a reduction in the WER by 71% in the case of a single vs. two sensors, and 78% in the case of a single vs. three sensors. The STAN models dramatically outperform

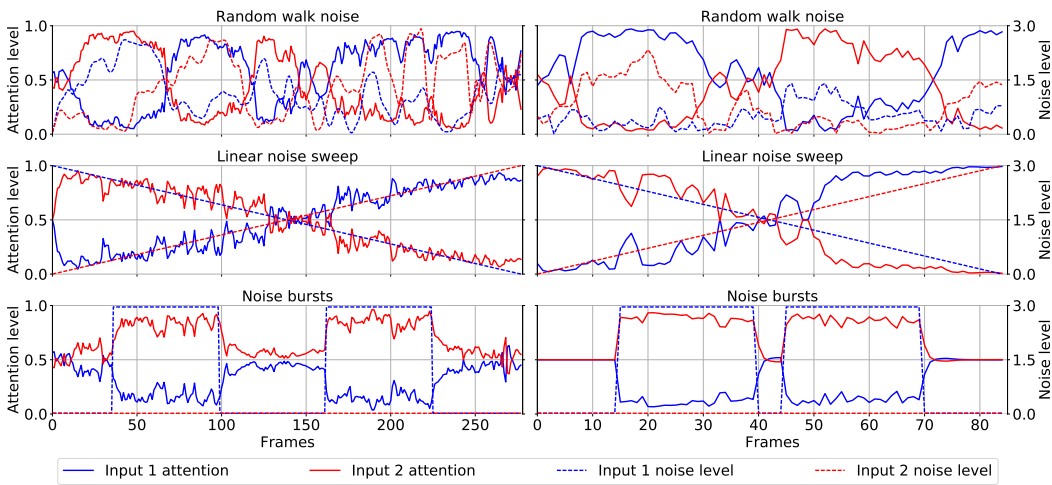

Figure 2: Attention response of a two-sensor STAN model trained on audio (TIDIGITS, left) and video (GRID, right) to random walk noise (top), linear noise sweeps (middle) and noise bursts (bottom). Both STANs show the desired negative correlation between noise level and sensor attention.

Table 2: Models for the GRID evaluation, with 'G'=GRU, 'bG'=bidirect. GRU and 'D'= dense unit

| Name | Architecture | Video sensors | Transformation Module | Attention Module | Classification Module | Parameters |
|---|---|---|---|---|---|---|
| Single Video | Baseline | 1 | CNN | None | (200,200,52)-bG,bG,D | 1.06M |
| Double Video STAN | STAN | 2 | CNN | (150,1)-G,D | (200,200,52)-bG,bG,D | 1.09M |

the concatenation models with an equivalent number of sensors, achieving around half the WER. This suggests that concatenation models had difficulties in prioritizing signal sources with lower noise levels.

## 3.2 SYNTHETIC NOISE ON VIDEO

**Dataset**    The GRID (Cooke et al., 2006) corpus is used for perceptual studies of speech processing. There are 1000 sentences spoken by each of the 34 speakers. The GRID word vocabulary contains four commands, four colors, four prepositions, 25 letters, ten digits, and four adverbs, resulting in 51 classes. Each sentence contains 6 vocabulary units. The video recordings were processed with the Dlib face detector to extract the faces and were then resized to 48 x 48 pixels per frame. The RGB channels in the frames were then normalized to zero-mean and unit-variance on the whole data set. As for TIDIGITS, the WER is used as a performance metric.

**Training**    The video sequences of the GRID database allow for a sequence-to-sequence mapping task, so the CTC objective was adopted. All models were trained with the ADAM optimizer for a maximum of 100 epochs, with early stopping preventing overfitting.

**Models**    Two video models as described in Table 2 are evaluated. Both use a CNN for feature transformation: three convolutional layers of 5x5x8 (5x5 filter size, 8 features), each followed by a 2x2 max pooling layer. The output is flattened and presented to the classification module. The STAN attention modules use a SELU non-linearity (Klambauer et al., 2017) on the dense unit (other nonlinearities such as leaky ReLU were found to work, too). The STAN model shares the parameters of the transformation and attention modules across both its sensors.

**Results**    The testing is carried out on the GRID test set, both on a clean variant and a noisy variant corrupted by random walk noise. The attention response is plotted in Figure 2(right). The STAN model shows proper functioning of the attentional mechanism for video data on random walk noise and robust generalization to linear sweep noise as well as burst noise. The WER for both the clean and noisy test set is reported in Figure 3(b). On the clean test set, STANs perform better with the

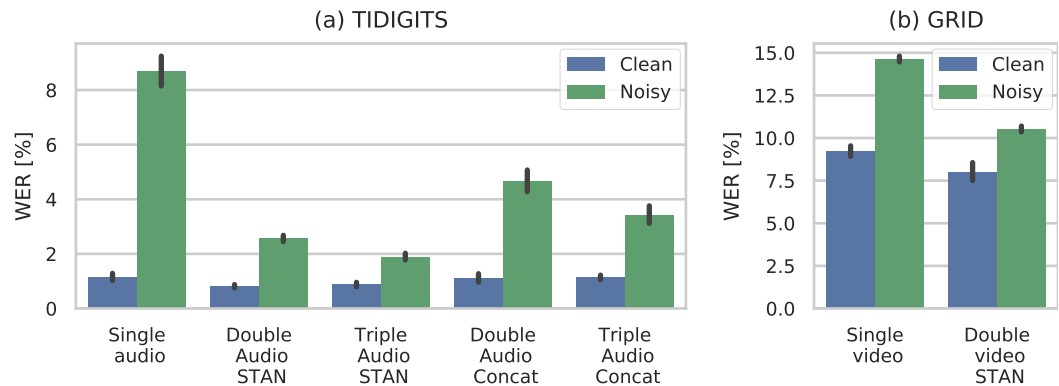

Figure 3: WER on (a) TIDIGITS and (b) GRID for clean and noisy test sets. The TIDIGITS results are based on 5 parameter initializations, while the GRID results are based on 4 parameter initializations.

WER reduced by relatively 13%. On the noisy test set, the STAN model performs significantly better and reduces the WER by relatively 28%, a clear indication of the effectiveness of the attentional mechanism.

## 4 MULTI-CHANNEL SPEECH RECOGNITION WITH NATURAL NOISE

**Dataset**   In a final experiment, STANs are evaluated on the CHiME-3 corpus (Barker et al., 2015), which allows for a *multi-channel* automatic speech recognition (ASR) experiment with *real-world* noisy speech. The corpus provides real and simulated noisy data from four environments: a cafe (CAF), a street junction (STR), public transport (BUS) and a pedestrian area (PED). The noisy speech data consists of 6-channel recordings of sentences from the WSJ0 corpus (Garofalo et al., 2007) spoken in the four environments. For recording, a tablet device with 6 microphones was used, with 5 microphones facing towards the speaker and 1 microphone facing away from the speaker (backward channel). The simulated data is also multi-channel and was constructed by mixing clean samples of WSJ0 with environment background recordings.

For training, both real (1600 samples, 'tr05_real') and simulated noisy speech data (7138 samples, 'tr05_simu') was used. For testing, real noisy speech data ('et05_real') was used in order to evaluate STANs on natural noise. The samples were preprocessed into 123-dimensional filterbank features (40 Mel-spaced filterbanks, energy coefficent, 1st and 2nd order delta features, 25ms frames, 10ms frame shift) and normalized to zero-mean and unit variance per sample.

**Models**   We compare two STAN variants against one sensor concatenation model. Both STAN variants use 6 sensors (one for each microphone channel) and identity transformation modules. Each sensor has an attention module consisting of 20 LSTM units followed by 1 dense unit with a SELU non-linearity (an arbitrary choice, as leaky ReLUs worked as well). The parameters of the attention modules are either shared across sensors (STAN-shared) or not shared across sensors (STAN-default). The concatenation model concatenates all 6 input sensors into a 738-dimensional feature representation. For classification, both STAN variants and the concatenation model use 4 layers of bidirectional LSTMs units with 350 units in each direction, followed by an affine transform to the 59 output classes (characters + blank label). The network output is decoded with a trigram language model based on Weighted Finite State Transducers (WFSTs) (Mohri et al., 2008) as described by Miao et al. (2015). We compare our models against the CHiME-3 official DNN/HMM hybrid baseline model (Weng et al., 2014) that uses 27M parameters (twice as much as our models) and a more complex training procedure (phoneme labels, forced alignment with GMM/HMM, maximum likelihood linear transform (MLLT), feature-space maximum likelihood linear regression (fMLLR) transform, state-level minimum Bayes risk (sMBR) criterion).

**Training**   The CTC objective was used to automatically learn the alignments between speech frames and label sequences. All models are trained with the ADAM optimizer for 150 epochs, selecting

Table 3: WER [%] in environments BUS, CAF, PED, STR and by average on 'et05_real', CHiME-3. The best result of an environment is printed bold.

| Model | BUS | CAF | PED | STR | Ave. | Parameters |
|---|---|---|---|---|---|---|
| DNN/HMM hybrid | 51.8 | 34.7 | **27.2** | **20.1** | 33.4 | 27.01M |
| STAN-default | **41.5** | **33.4** | 28.4 | 22.6 | **31.5** | 13.22M |
| STAN-shared | 43.4 | 33.7 | 28.8 | 22.5 | 32.1 | 13.17M |
| Concatenation | 43.4 | 33.6 | 30.9 | 24.5 | 33.1 | 14.94M |

the model with lowest WER for evaluation. For regularization, Gaussian noise on the inputs ($\mu = 0$, $\sigma = 0.6$), dropout (p=0.3) and weight decay (1e-4) were applied.

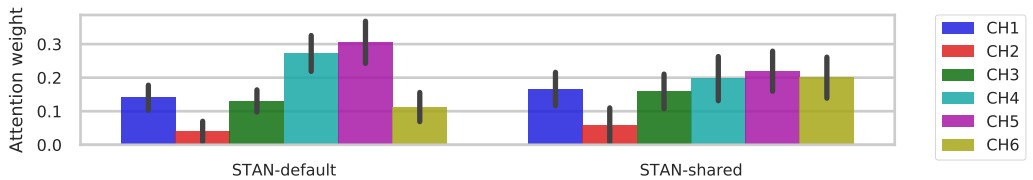

Figure 4: Attention weights per channel averaged over all environments on 'et05_real', CHiME-3. Both STAN-variants attribute the lowest attention to channel 2 (backward channel). For channels 1/3/4/5/6, STAN-shared spreads attention more equally than STAN-default, which appears to prefer channels 4 and 5.

**Results** As reported in Table 3, all of our CTC models perform better than the baseline DNN/HMM hybrid in terms of average WER by relatively 0.9% (concatenation model), 3.9% (STAN-shared) and 5.9% (STAN-default). Both STAN variants outperform the concatenation model. For further analysis, we first verify the function of the STAN attention modules and come back to the comparison against concatenation models later.

**Attentional mechanism on natural noise** We first report on the average attention weight for each channel over the whole evaluation set 'et05_real' in Figure 4. We recall that channel 2 faces away from the speaker and generally has a lower SNR than the other channels. On average, both STAN variants attribute the lowest weight to channel 2. This result demonstrates two key features of STANs: firstly, STANs are able to tune their attention towards more useful sensors even on real-world noisy data. Secondly, the output of the attention modules is highly informative, clearly indicating a sub-optimal sensor. Avoiding channel 2 seems to be the easier task for the variant STAN-default, as the channel 2 attention module could be learned in a way that it constantly outputs lower attention weights. Remarkably, STAN-shared is able to differentiate between channel 2 and the remaining channels. Within the shared parameter space of its attention modules, this seems a harder task than for STAN-default, as the shared attention module must learn to simultaneously compute a high attention score on a sensor with high SNR and a low attention score on a sensor with low SNR, even in the presence of natural noise. For the front-facing channels (1/3/4/5/6) STAN-shared attributes similar attention weights, while STAN-default seems to prefer channels 4 and 5. We plot the 6 channels and attention weights for a sample that suffers channel corruption on multiple channels (1/2/4) in in Figure 5. By looking at the attention signals of this sample, we observe that both STANs are able to tune dynamically the attention level of each sensor, as depicted by the lowering of attention on a sensor that is temporally corrupted (channel 4 after frame 120) and the continuous suppression of the other corrupted channels 1 and 2. The STAN-shared attention weights seems generally more interpretable than those of the STAN-default. Five more plots are shown in Appendix B, covering samples with various configurations of corrupted channels and including the merged representation.

**Effectiveness of attentional mechanism** With the ability of STANs to reduce attention on corrupted channels, the question remains why STANs only achieved 3.0% to 4.9% lower WER than the concatenation model. This is explained by a closer look at the CHiME-3 dataset. Recall that we trained our models on simulated data (7138 samples) and real data (1600) samples. The CHiME authors state that 12% of the real recordings suffer channel corruption (hardware issues, masking by

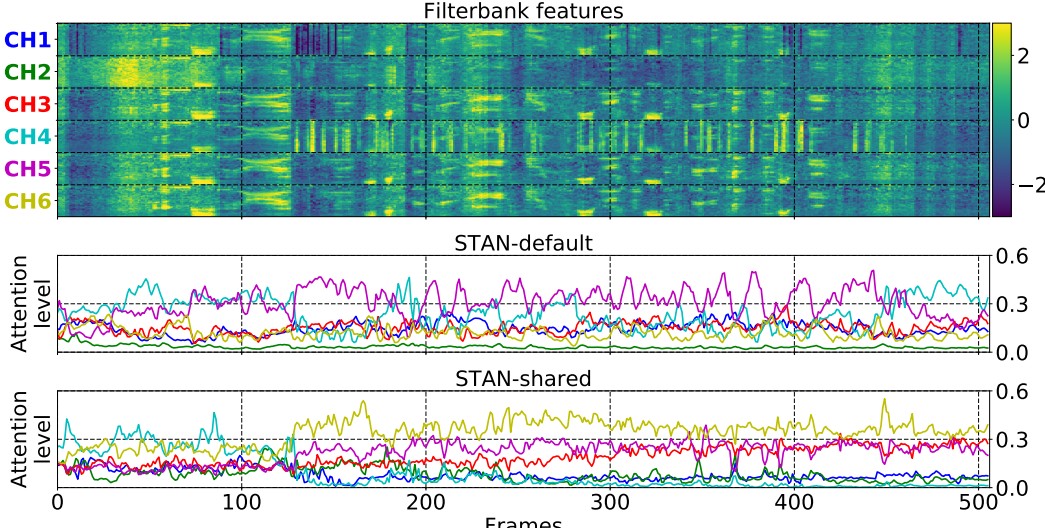

Figure 5: Filter bank features (top) of the sample 'M05_443C020Q_BUS' from CHiME-3 and attention response of STAN-default (middle) and STAN-shared (bottom). The channels are color-coded. For better visibility of the channel differences, the features plot has been clipped to the range -3 to 3. The attention generally follows the signal quality, with clear suppression of the attention on the corrupted channels 1, 2 and 4. Note how the attention on channel 4 is initially high, but then suppressed when the channel is temporarily corrupted after frame 120. The attention response of STAN-shared is more interpretable.

hands or clothes etc.). With such a small portion of corrupted samples, a standard model without sensory attention (i.e. the concatenation model) is still expected to perform well overall. To test this hypothesis, we assess the performance as a function of the corruption of samples. The CHiME authors provided a table ('mic_error.csv') where for each real sample, the cross correlation coefficients of all 6 channels relative to the reference close-talk microphone is given. Based on this table, we computed for each sample, the standard deviation across the cross correlation coefficients. A high standard deviation corresponds to a high likelihood of at least one channel being different (i.e. corrupted), which allows us to establish a ranking of potentially corrupted samples. The ranking was verified by listening tests and is considered as a solid indicator of corrupted samples. As a metric, we use the *partial* $WER_i$, which is the WER including the $i$ highest ranked samples in our corruption ranking. The results are shown for each of the 'et05_real' environments in Figure 6, with the $WER_i$ of STANs given relative to the concatenation models. We find that, for a higher share of corrupted samples (i.e. fewer samples included), STANs perform significantly better than concatenation models. When including the 50 most corrupted test samples of each environment, the $WER_{50}$ is reduced relatively by 12% (STAN-default) and 9% (STAN-shared). When looking at single environments, the relative $WER_{50}$ reduction grows larger on STR (23%, STAN-default) and PED (14%, STAN-shared). On the CAF and BUS environments, STANs still perform better, but to lesser extent. Samples from the latter two subsets are generally less susceptible to channel corruption, as these environments seem more controlled (presumably the speaker is seated and has better grip of the tablet without masking, no wind blows etc.).

**Robustness to channel removal** Due to their modular architecture, STANs are highly flexible with respect to sensor configurations. This is demonstrated by a channel removal experiment, where between one to five channels are removed on a STAN trained with all 6 channels. After removal, no additional training is allowed. Therefore, the sensor merge module only sees the remaining, active channels. Note that we do not zero out channels, but rather remove them from the computation graph. Such a flexibility is much harder to achieve for a concatenation model, as a disabled channel would change the input size of the network.

The results are reported in Table 4. In the first phase, one channel is removed at a time. For all channels except channel 2, the WER increases by relatively up to 5.7% (channel 5 on STAN-default). Note that channel 5 was the preferred channel of STAN-default, so even when removing the preferred channel

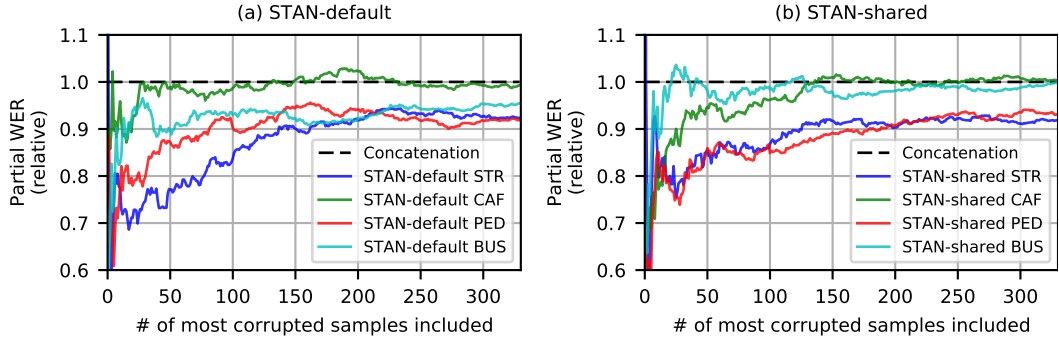

Figure 6: Partial WER computed up to the number of most corrupted samples for (a) STAN-default and (b) STAN-shared. The WER is given relative to the concatenation model. The advantage of STANs is especially large if samples suffer channel corruption.

Table 4: WER [%] when removing a single channel or multiple channels at a time on CHiME-3

| Model | Base | Single channel removed | | | | | | Multiple channels removed | | | |
| | | CH1 | CH2 | CH3 | CH4 | CH5 | CH6 | CH1/2 | CH1/2/3 | CH1/2/3/4 | CH1/2/3/4/5 |
|---|---|---|---|---|---|---|---|---|---|---|---|
| STAN-default | 31.5 | 32.1 | 30.9 | 31.9 | 32.6 | 33.3 | 32 | 31.6 | 32.2 | 34.1 | 39.7 |
| STAN-shared | 32.1 | 32.5 | 31.2 | 32.4 | 33 | 33.3 | 33.2 | 31.8 | 32.3 | 33.9 | 39.9 |

of a STAN variant, the model seems capable of exploiting the remaining channels with acceptable performance. Removing channel 2 (the backward channel) decreased the WER by relatively 2% on both STAN variants. In a second phase, multiple channels are removed in a sequential manner, starting with channels 1/2. For up to three removed channels (CH1/2/3), the WER remains stable within 2% of the 6-channel STANs. With five removed channels (CH1/2/3/4/5), the WER increases relatively by up to 26%. While this is a clear deterioration of performance, the performance still does not fully collapse. When removing sensors, we observed that the standard deviation of the merged representation increased with the number of removed sensors from around $\sigma = 0.85$ (all channels active) to $\sigma = 1$ (one channel active), which could push the network out of its preferred operating range and consequently cause the performance loss.

## 5 DISCUSSION

The sensor transformation attention network (STAN) architecture has a number of interesting features for sensor selection which we explored in this work. By equipping each sensor with an attentional mechanism for distinguishing meaningful features, networks can learn how to select, transform, and interpret the output of their sensors. Firstly, and by design, we show that STANs exhibit remarkable robustness to both real-world and synthetic dynamic noise sources. By challenging these networks during training with dynamic and persistent noise sources, the networks learn to rapidly isolate sensors corrupted by noise sources. Secondly, we show that this form of training results in even better accuracy performance from STANs than simply concatenating the sensor inputs. This is best demonstrated on the heavily noise corrupted STR environment of the CHiME-3 real-data evaluation set, where STANs achieve 23% lower WER than concatenation models for the 50 most corrupted samples. Thirdly, we find that the output of the attention modules is highly informative, clearly indicating a sub-optimal sensor placement for a sensor pointing away from the speaker on the CHiME-3 dataset. Remarkably, this outcome is even obtained when sharing the weights of the attention modules across sensors, implying that these attention modules learned to successfully differentiate between sensors with higher and lower SNR data in presence of natural noise.

Due to their modular architecture, STANs are also remarkably flexible with respect to the sensor configuration, even performing well with the removal of sensors after training. One can train STANs to solve a task with a multi-sensor setup and after training, remove the less informative sensors, with possibly savings of energy consumption and computational load on multi-sensor hardware systems with restricted computational power such as mobile robots. In the case of a defect, a sensor could be removed and STANs would remain operational with the remaining sensors.

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

## A    RANDOM WALK NOISE

In order to encourage the learning of the STAN attentional mechanism, a unique Gaussian noise source with varying noise level over time is added to each sensor. We refer to this noise type as *random walk noise*. The noise model aims to have an approximately uniform coverage of noise level over a range $[0, \sigma_{max})$ and no settle-in time that could introduce a sequence length dependence on the noise. The standard deviation of the noise $\sigma$ for an input sequence of $t$ timesteps can be calculated thusly:

$$\sigma(t) = \phi \left( \underbrace{\sigma_0 + \sum_{i=1}^{t} \text{sgn}(s_i)n_i,}_{a} \quad \sigma_{max} \right),$$

$$\sigma_0 \sim \mathcal{U}(0, \sigma_{max}/2), \quad s_i \sim \mathcal{U}(-1, 1) \quad n_i \sim \Gamma(k, \theta)$$

(3)

with $\sigma_0$ drawn uniformly over the range $[0, \sigma_{max}/2)$ and $n_i$ drawn from a gamma distribution with shape $k$ and scale $\theta$. The signum function extracts positive and negative signs from $s_i$ with equal probability. A parameter search during our experiments yielded $\sigma_{max} = 3$, $k = 0.8$ and $\theta = 0.2$ as an appropriate set of parameters. We define the reflection function $\phi(a, \sigma_{max})$ as

$$\phi(a, \sigma_{max}) = \sigma_{max} - \left| \text{mod}(a, 2\sigma_{max}) - \sigma_{max} \right|$$

(4)

where $\text{mod}(a, 2\sigma_{max}) = a - 2\sigma_{max} \lfloor a/2\sigma_{max} \rfloor$ maintains the values within the desired range $(0, 2\sigma_{max})$ and the subsequent shift and magnitude operations map the values to the range $[0, \sigma_{max})$ while avoiding discontinuities. Finally the input data $x$ at feature index $k$ and time index $t$ is mixed with the noise sampled from a normal distribution as follows:

$$x_{k,t} = x_{k,t} + n_{k,t}, \qquad n_{k,t} \sim \mathcal{N}(0, \sigma^2(t))$$

(5)

The reflection function $\phi(a, \sigma_{max})$ performs similarly to the $\text{mod}$ operator, but at the edges, produces a continuous reflection about the edge instead of a discontinuous wrap. Therefore, this forms a constrained random walk, limited by $\sigma_{max}$, which will become the standard deviation of normally distributed random noise added to the input $x$ at feature index $k$ and time point $t$. This noise model generates sequences that provide a useful training ground to tune the attentional mechanism of STAN models, as the noise level varies over time and allows periods of low noise (high attention desired) and high noise (low attention desired). An example for video frames is shown in Figure 7.

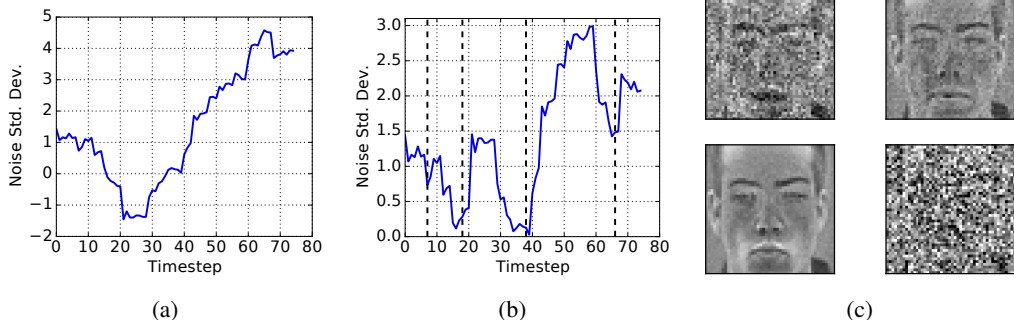

| (a) | (b) | (c) |

Figure 7: Depiction of the random walk noise added during training. In **(a)**, the cumulative sum of a sequence of random variables forms a random walk. In **(b)**, the random walk becomes bounded after applying the reflection operator $\phi$ in Eq. 4. On the four sub-panels in **(c)**, a visualization of the noise drawn at each time point. Each panel depicts a video frame from the GRID corpus, zero mean and unit variance normalized, mixed with a Gaussian noise source whose standard deviation corresponds to a vertical dotted line in **(b)**.

## B SAMPLES FROM THE CHiME-3 DATABASE

Followingly, we plot 6 samples from the 'et05_real' evaluation set. The samples are summarized in Table 5, with corrupted channels given after visual and listening inspection. The samples were chosen in a way that every channel was corrupted in at least one of the samples, and also to include one sample where there is minimal corruption. The figures show the filterbank features, the attention response of STAN-shared and STAN-default and the output of the sensor merge module that is seen by the classification module. We recall that all these samples are part of the *real* evaluation set which contains data with natural noise. The filterbank features and merge layer images are clipped to the range -3 to 3 to make differences more visible. Viewing on a high-resolution screen or a high-quality printout is highly recommended, additionally, we recommend listening to the audio samples if the reader is in possession of the dataset.

Table 5: Sample keys and corrupted channels (based on visual inspection and listening tests), ordered by the number of corrupted channels. All samples are from the 'et05_real' evaluation set, CHiME-3

| Sample key | Corrupted channels | Figure |
|---|---|---|
| M06_440C0209_CAF | None | Figure 8 |
| M06_443C020P_BUS | 2 | Figure 9 |
| F06_446C0210_BUS | 5 | Figure 10 |
| F06_442C020N_STR | 2/6 | Figure 11 |
| M05_440C2010_PED | 2/3/4 | Figure 12 |
| M05_443C020Q_BUS | 1/2/4 | Figure 13 |

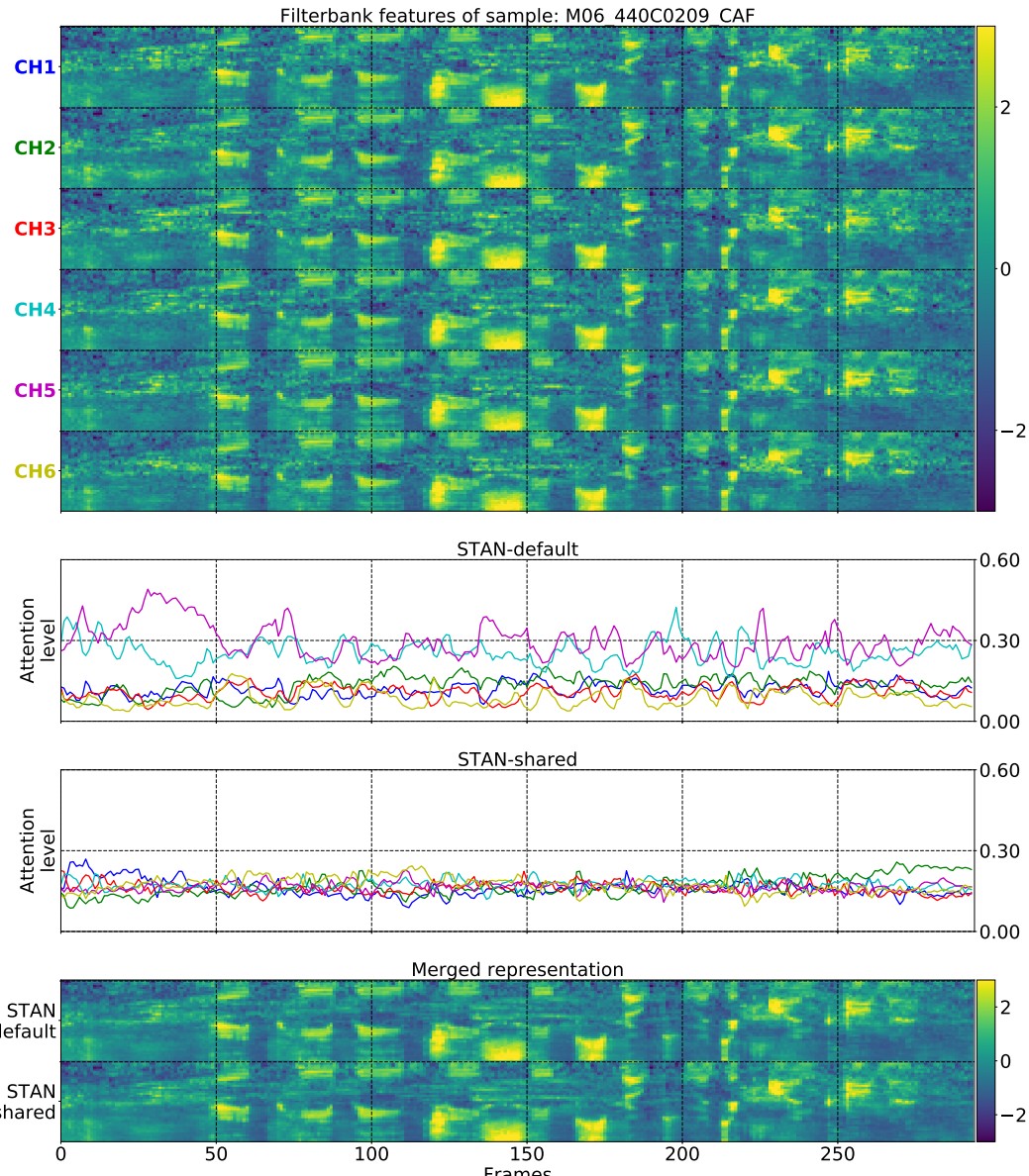

Figure 8: Corrupted channels - none. This sample shows the native attention response when no channel is corrupted, that could otherwise be interpreted as the bias towards channels. STAN-shared seems to express no channel preference, while STAN-default prefers channels ch4 and ch5. The merged representations appear smoother than the single channels.

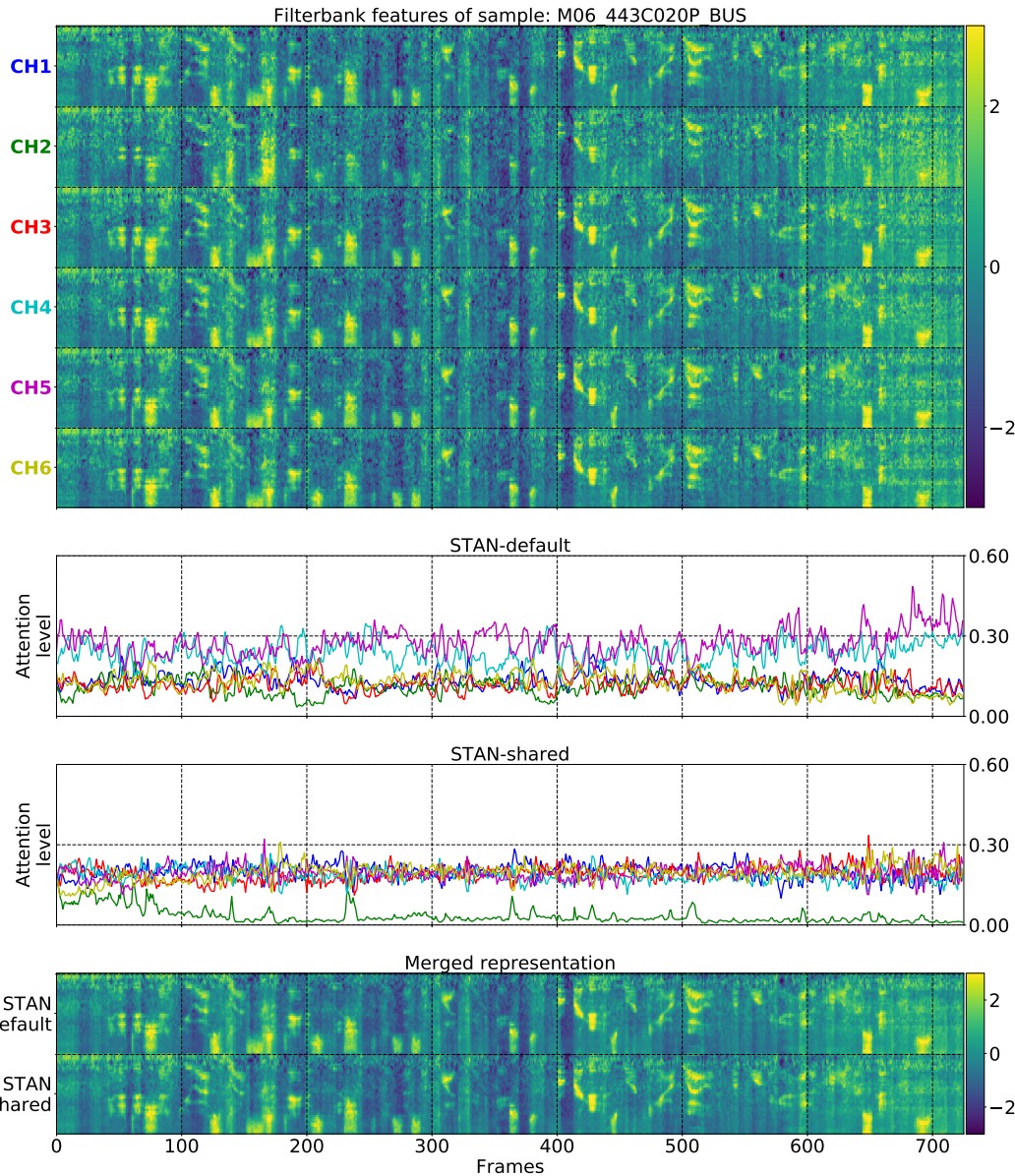

Figure 9: Corrupted channels - ch2. This sample is representative for most of the real evaluation set: the backward channel 2 is slightly corrupted, while the other channels seem similar. Remarkably, STAN-shared is able to detect the backward channel although the attention module weights are shared across channels. It seems that the STAN-shared attention modules are able to simultaneously compute high attention scores on channels with high SNR and low attention scores on channels with low SNR, even in the presence of natural noise.

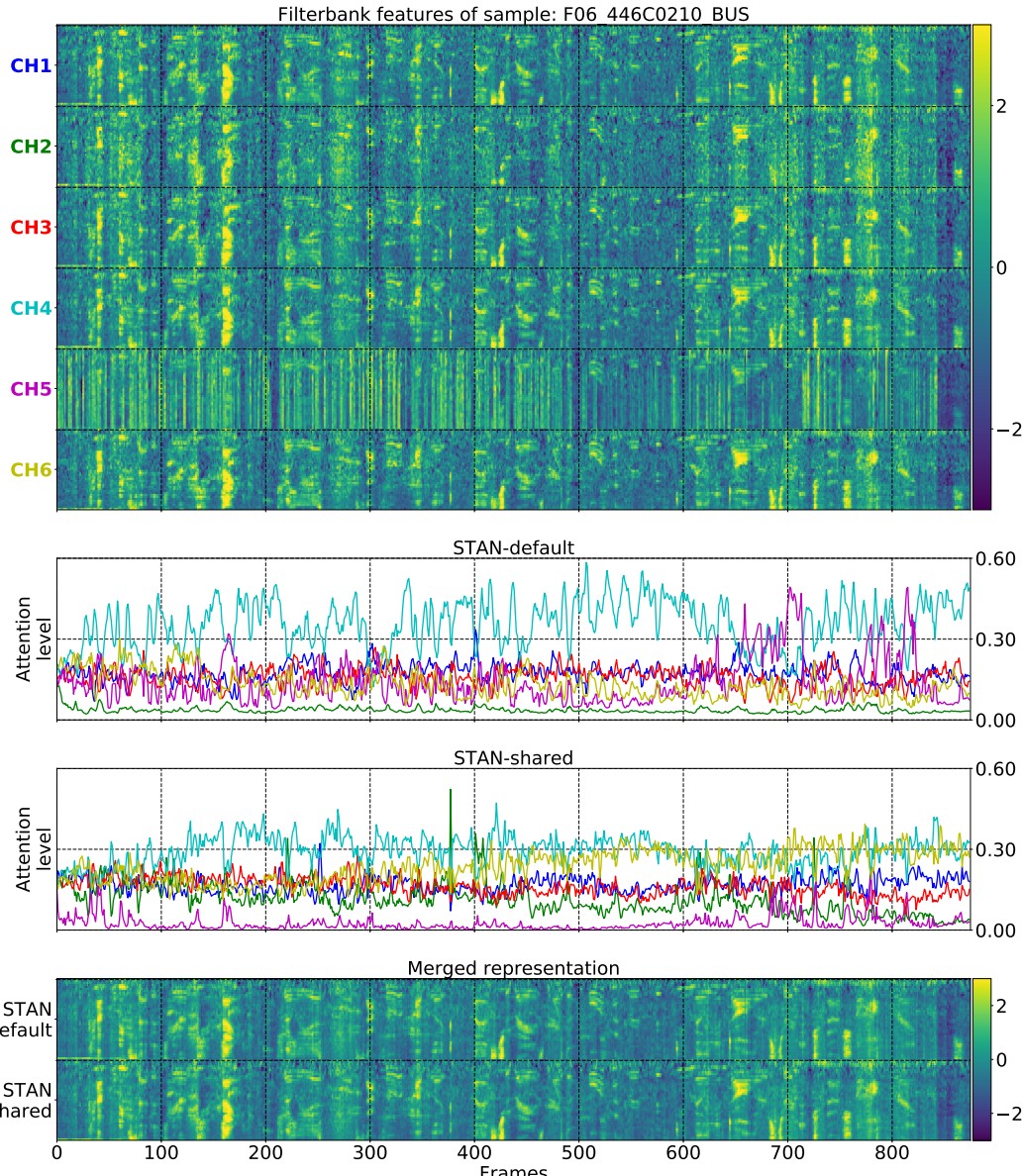

Figure 10: Corrupted channels - ch5. Even though the preferred channel of STAN-default is corrupted, the attentional mechanism is able to tune in on the other channels, especially channel 4. The merged representations appear unaffected by the heavily corrupted channel 5.

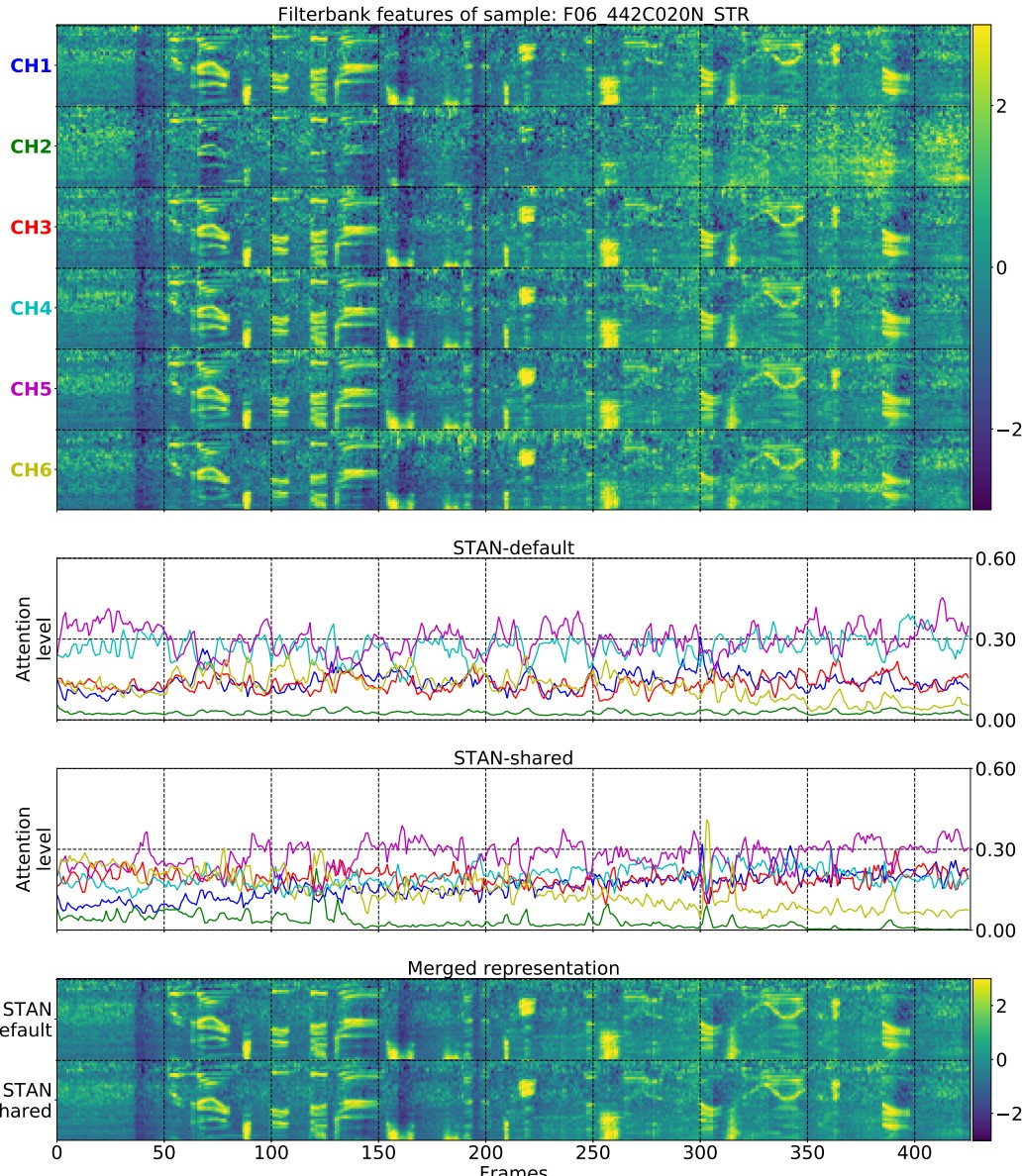

Figure 11: Corrupted channels - ch2/6. Both STAN variants reduce their attention towards channel 6 at the end of the sequence, where channel 6 seems most corrupted. Channel 2 is suppressed over the whole sequence.

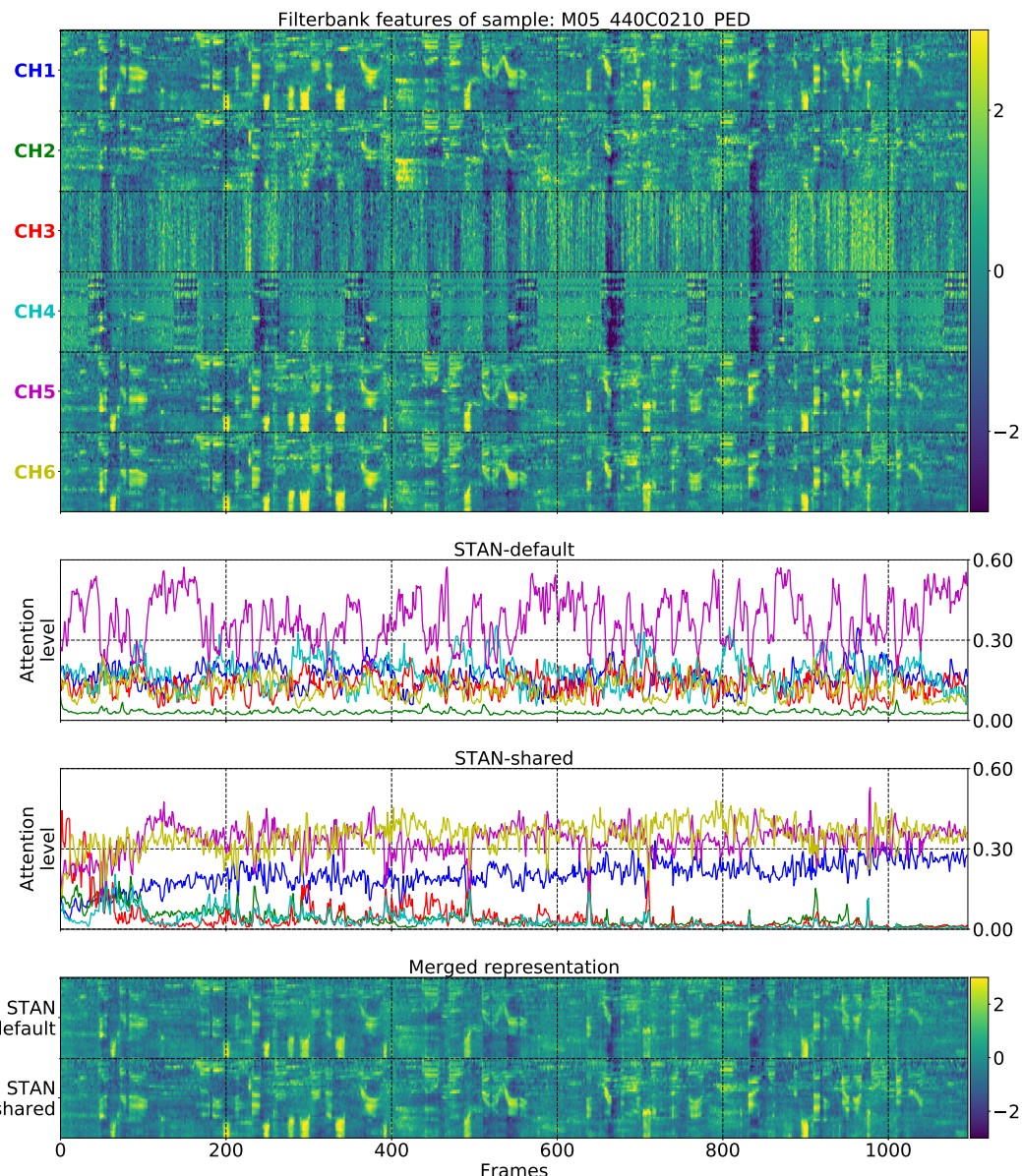

Figure 12: Corrupted channels - ch2/3/4. The attention roughly follows the signal quality, with clear suppression of the attention on the corrupted channels 2, 3 and 4. The attention response of STAN-shared is more interpretable. The merged representations are less noisy than the single channels, while still resolving fine details as between frames 550 and 600.

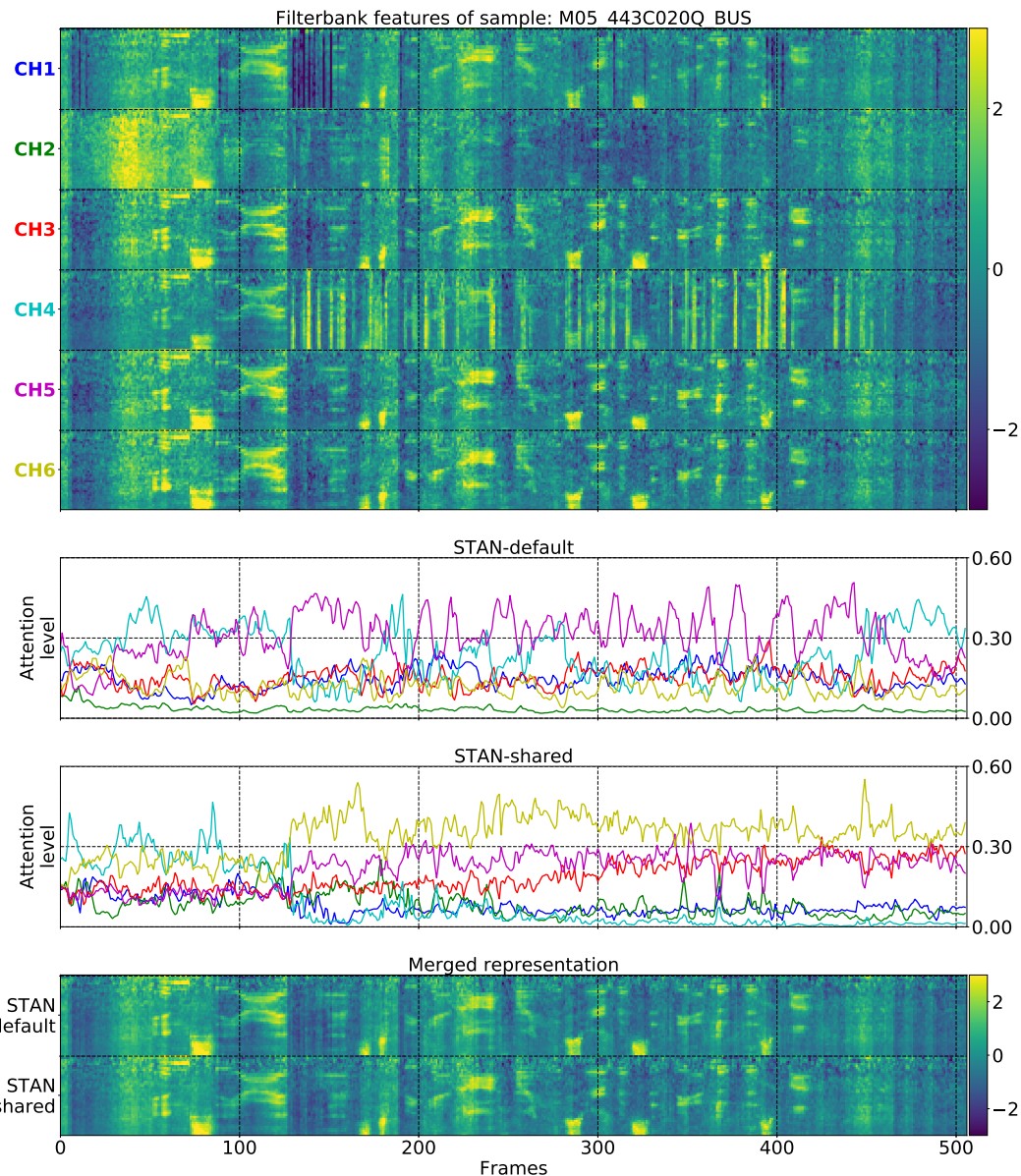

Figure 13: Corrupted channels - ch1/2/4. This is the plot from the main section (Figure 5). The attention roughly follows the signal quality, with clear suppression of the attention on the corrupted channels 1, 2 and 4. Note how the attention on channel 4 is initially high, but then suppressed when the channel is temporarily corrupted after frame 120. The attention response of STAN-shared is more interpretable.

