# OpenReview forum: "Sensor Transformation Attention Networks"
_ICLR.cc/2018/Conference — Reject_

### Official Review · AnonReviewer1 · 2017-11-27
**Interesting analysis for real noisy multichannel scenarios.**

**Rating:** 7
**Confidence:** 4

**Review:**

This paper proposes sensor transformation attention network (STAN), which dynamically select appropriate sequential sensor inputs based on an attention mechanism.

Pros:
One of the main focuses of this paper is to apply this method to a real task, multichannel speech recognition based on CHiME-3, by providing its reasonable sensor selection function in real data especially to avoid audio data corruptions. This analysis is quite intuitive, and also shows the effectiveness of the proposed method in this practical setup.

Cons:
The idea seems to be simple and does not have significant originality. Also, the paper does not clearly mention the attention mechanism part, and needs some improvement.

Comments:
-	The paper mainly focuses on the soft sensor selection. However, in an array signal processing context (and its application to multichannel speech recognition), it would be better to mention beamforming techniques, where the compensation of the delays of sensors is quite important.
-	In addition, there is a related study of using multichannel speech recognition based on sequence-to-sequence modeling and attention mechanism by Ochiai et al, "A Unified Architecture for Multichannel End-to-End Speech Recognition with Neural Beamforming," IEEE Journal of Selected Topics in Signal Processing. This paper uses the same CHiME-3 database, and also showing a similar analysis of channel selection. It’s better to discuss about this paper as well as a reference.
-	Section 2: better to explain about how to obtain attention scores z in more details.
-	Figure 3, experiments of Double audio/video clean conditions: I cannot understand why they are improved from single audio/video clean conditions. Need some explanations.
-	Section 3.1: 39-dimensional Mel-frequency cepstral coefficients (MFCCs) -> 13 -dimensional Mel-frequency cepstral coefficients (MFCCs) with 1st and 2nd order delta features.
-	Section 3.2 Dataset “As for TIDIGIT”: “As for GRID”(?)
-	Section 4 Models “The parameters of the attention modules are either shared across sensors (STAN-shared) or not shared across sensors (STAN- default).”: It’s better to explain this part in more details, possibly with some equations. It is hard to understand the difference.

---

### Official Review · AnonReviewer3 · 2017-11-27
**Intuitive idea, lack of comparison.**

**Rating:** 4
**Confidence:** 4

**Review:**

The manuscript introduces the sensor transformation attention networks, a generic neural architecture able to learn the attention that must be payed to different input channels (sensors) depending on the relative quality of each sensor with respect to the others. Speech recognition experiments on synthetic noise on audio and video, as well as real data are shown.

First of all, I was surprised on the short length of the discussion on the state-of-the-art. Attention models are well known and methods to merge information from multiple sensors also (very easily, Multiple Kernel Learning, but many others).

Second, from a purely methodological point of view, STANs boil down to learn the optimal linear combination of the input sensors. There is nothing wrong about this, but perhaps other more complex (non-linear) models to combine data could lead to more robust learning.

Third, the experiments with synthetic noise are significant to a reduced extend. Indeed, adding Gaussian noise to a replicated input is too artificial to be meaningful. The network is basically learning to discard the sensor when the local standard deviation is high. But this is not the kind of noise found in many applications, and this is clearly shown in the performances on real data (not always improving w.r.t state of the art). The interesting part of these experiments is that the noise is not stationary, and this is quite characteristic of real-world applications. Also, to be fair when discussion the results, the authors should say that simple concatenation outperforms the single sensor paradigm.

I am also surprised about the baseline choice. The authors propose a way to merge/discard sensors, and there is no comparison with other ways of doing it (apart from the trivial sensor concatenation). It is difficult to understand the benefit of this technique if no other baseline is benchmarked. This mitigates the impact of the manuscript.

I am not sure that the discussion in page corresponds to the actual number on Table 3, I did not understand what the authors wrote.

---

### Official Review · AnonReviewer2 · 2017-11-27
**Review: "Sensor Transformation Attention Networks"**

**Rating:** 3
**Confidence:** 4

**Review:**

Summary:

The authors consider the use of attention for sensor, or channel, selection. The idea is tested on several speech recognition datasets, including TIDIGITS and CHiME3, where the attention is over audio channels, and GRID, where the attention is over video channels. Results on TIDIGITS and GRID show a clear benefit of attention (called STAN here) over concatenation of features. The results on CHiME3 show gain over the CHiME3 baseline in channel-corrupted data.

Review:

The paper reads well, but as a standard application of attention lacks novelty. The authors mention that related work is generalized but fail to differentiate their work relative to even the cited references (Kim & Lane, 2016; Hori et al., 2017). Furthermore, while their approach is sold as a general sensor fusion technique, most of their experimentation is on microphone arrays with attention directly over magnitude-based input features, which cannot utilize the most important feature for signal separation using microphone arrays---signal phase. Their results on CHiME3 are terrible: the baseline CHiME3 system is very weak, and their system is only slightly better! The winning system has a WER of only 5.8%(vs. 33.4% for the baseline system), while more than half of the submissions to the challenge were able to cut the WER of the baseline system in half or better! http://spandh.dcs.shef.ac.uk/chime_challenge/chime2015/results.html. Their results wrt channel corruption on CHiME3, on the other hand, are reasonable, because the model matches the problem being addressed…

Overall Assessment:

In summary, the paper lacks novelty wrt technique, and as an “application-of-attention” paper fails to be even close to competitive with the state-of-the-art approaches on the problems being addressed. As such, I recommend that the paper be rejected.


Additional comments:

-	The experiments in general lack sufficient detail: Were the attention masks trained supervised or unsupervised? Were the baselines with concatenated features optimized independently? Why is there no multi-channel baseline for the GRID results?
-	Issue with noise bursts plot (Input 1+2 attention does not sum to 1)
-	A concatenation based model can handle a variable #inputs: it just needs to be trained/normalized properly during test (i.e. like dropout)…

---

### Author Response · Authors · 2018-01-05
**General comments to reviewers**

We thank the reviewers for their time in reviewing the submission. Our omission on the specific comparisons of our work to other systems such as Kim et al, and Hori et al, was unintended and will be corrected in the updated manuscript.
We realize that we are unable to change the document significantly at this time and will take the reviewers comments into consideration when we write our next revision.

---

### Decision · Program_Chairs · 2018-01-29
**ICLR 2018 Conference Acceptance Decision**

**Decision:**

Reject

**Comment:**

Meta-score: 4

This paper presents an approach which uses attention across multiple speech or video channels.  After some synthetic experiments, presents experiments on chime-3, but has a rather weak baseline system

Pros:
 - addresses an interesting task

Cons:
 - does not take account of other recent papers in the area
 - experimental results are weak - very high errors in baseline system
 - limited novelty